# Atomically resolved imaging of the conformations and adsorption geometries of individual β-cyclodextrins with non-contact AFM

**Márkó Grabarics** [1,2,8], **Benjamín Mallada** [3,4,8], **Shayan Edalatmanesh** [3,4,8], **Alejandro Jiménez-Martín** [3,4,5], **Martin Pykal** [4], **Martin Ondráček** [3], **Petra Kührová** [4], **Weston B. Struwe** [2,6], **Pavel Banáš** [4] ✉, **Stephan Rauschenbach** [1,2] ✉, **Pavel Jelínek** [3,4] ✉ & **Bruno de la Torre** [4,7] ✉

Glycans, consisting of covalently linked sugar units, are a major class of biopolymers essential to all known living organisms. To better understand their biological functions and further applications in fields from biomedicine to materials science, detailed knowledge of their structure is essential. However, due to the extraordinary complexity and conformational flexibility of glycans, state-of-the-art glycan analysis methods often fail to provide structural information with atomic precision. Here, we combine electrospray deposition in ultra-high vacuum with non-contact atomic force microscopy and theoretical calculations to unravel the structure of β-cyclodextrin, a cyclic glucose oligomer, with atomic-scale detail. Our results, established on the single-molecule level, reveal the different adsorption geometries and conformations of β-cyclodextrin. The position of individual hydroxy groups and the location of the stabilizing intramolecular H-bonds are deduced from atomically resolved images, enabling the unambiguous assignment of the molecular structure and demonstrating the potential of the method for glycan analysis.

Glycans are present in all cells, playing key roles in a range of biological processes and contributing greatly to the molecular and functional diversity of living organisms[1,2]. Structurally, they consist of monosaccharide building blocks, such as glucose, mannose, and galactose, which are connected to each other via their OH groups, forming glycosidic linkages. Much of the structural diversity of glycans stems from variations in the orientation of these OH groups. Such variations lead to the isomerism of monosaccharide units, and the variability in the

orientation of the glycosidic bonds between them. Combined with differences in linkage position, these small variations lead to the intrinsic and significant structural complexity of glycans. As minute structural differences can fine-tune biological function, analytical approaches sensitive to such small structural details and capable of revealing the underlying structure of glycans are highly sought after[3].

This immense complexity of glycans impedes complete structural assignments. Their elaborate regio- and stereochemistry pose

[1]Department of Chemistry, University of Oxford, OX1 3QU Oxford, UK. [2]Kavli Institute for Nanoscience Discovery, University of Oxford, OX1 3QU Oxford, UK. [3]Institute of Physics, Czech Academy of Sciences, 16200 Prague, Czech Republic. [4]Czech Advanced Technology and Research Institute, Palacký University Olomouc, 78371 Olomouc, Czech Republic. [5]Faculty of Nuclear Sciences and Physical Engineering, Czech Technical University in Prague, 115 19 Prague, Czech Republic. [6]Department of Biochemistry, University of Oxford, OX1 3QU Oxford, UK. [7]Nanomaterials and Nanotechnology Research Center, CSIC-UNIOVI-PA, 33940 El Entrego, Spain. [8]These authors contributed equally: Márkó Grabarics, Benjamín Mallada, Shayan Edalatmanesh. ✉e-mail: pavel.banas@upol.cz; stephan.rauschenbach@chem.ox.ac.uk; jelinekp@fzu.cz; bruno.de@upol.cz

significant challenges to state-of-the-art mass spectrometric methods[4,5]. Isomers with identical masses are difficult to distinguish as they usually display very similar mass spectrometric fragmentation patterns, and the de novo assignment of linkage positions is hindered by the scarcity of diagnostic cross-ring fragments. In addition, atomic-resolution imaging of glycans by ensemble-averaged microscopic and diffraction methods is often not possible due to their conformational flexibility. Consequently, our knowledge of the structure of glycans is often incomplete, which represents a major bottleneck in better understanding their functions and facilitating their applications, from vaccine and drug development to materials science[6–10].

An alternative approach to fully characterize the structure of glycans relies on the combination of electrospray ionization-based deposition methods and scanning tunneling microscopy (STM), which enables the gentle deposition of the thermolabile glycans onto clean single-crystal surfaces, and subsequent real-space imaging at the single-molecule level[11–16]. To obtain images with sufficient quality that allow for visualizing the connectivity of monosaccharide units within glycans, measurements must be performed under highly controlled conditions, requiring ultrahigh vacuum (UHV), cryogenic temperatures, and atomically flat surfaces. However, STM images obtained with bare metal tips reveal the overall shape of individual monosaccharide building blocks, which also depends on their adsorption conformation[11,17], while internal structural details at the atomic level remain unresolved.

Here we employ dynamic non-contact atomic force microscopy (nc-AFM) with CO-functionalized tips—enabled by electrospray deposition (ESD) of the molecules—to reveal the internal structure of monosaccharide units and resolve atomic-scale features that reflect individual functional groups and chemical bonds within the glycans. We investigate β-cyclodextrin (β-CD) as a model compound, a cyclic oligosaccharide consisting of seven D-glucopyranose units connected via α−1,4-glycosidic bonds (Fig. 1a). The shape of this macrocycle resembles a truncated cone where the glucose units enclose a central hydrophobic cavity[18]. Fourteen secondary OH groups form a wider rim at the cone's base (secondary face), while seven primary OH groups form a narrower rim at the top (primary face). Its three-dimensional shape renders β-CD a challenging molecule for nc-AFM and expands the scope of this technique, which has so far been applied mainly to conjugated planar molecular structures[19–23]. By combining experiments with nc-AFM simulations that also take electrostatic tip-molecule interactions into account, we show that the obtained images contain sufficient information to distinguish and unambiguously assign the different OH groups within single β-CD molecules, and to

confidently identify the atomic structure and conformation of this complex biomolecule.

## Results

### Single-molecule imaging of β-cyclodextrin conformers

Incompatible with conventionally used thermal sublimation methods, β-CD was deposited onto an atomically clean Au(111) single-crystal substrate at room temperature using ESD. Low-temperature STM topographs of the samples reveal irregular two-dimensional islands with average topographic heights between 250 and 350 pm (Fig. 1e and Supplementary Fig. 1). Within the islands, single β-CD molecules emerge as bright, doughnut-shaped objects. They consist of seven clearly resolved rounded lobes that encircle a central depression, which can be tentatively assigned to the individual glucose building blocks and the molecule's cavity, respectively. The intact β-CDs are surrounded by a lower bed that consists of distorted macrocycles (and potentially some smaller fragments), discussed later in more detail (additional images of the structures forming the bed are shown in Supplementary Fig. 2).

Although STM reveals overall shapes and monomer units, a much more precise assignment can be achieved using high-resolution nc-AFM. Primarily, three dominant types of objects are identified on the surface, all illustrated in Fig. 1b–d. The first type (Fig. 1b) exhibits sevenfold symmetry with an outside diameter of 12.6 ± 0.5 Å. It features a central heptagon with bright, distinct borders enclosing a darker interior. A narrow ray points outwards from each corner of the heptagon, with darker regions between them, resulting in a star-shaped structure. This structure is surrounded by a faint, diffuse halo that gradually dissipates into the substrate's background. The second type (Fig. 1c) appears larger across (15.6 ± 0.1 Å). Seven round features, separated by narrow ribs, are arranged in a circular pattern around a central cavity. The differences in their diameter (12.6 vs. 15.6 Å), along with their similar absorption height relative to the substrate (roughly 7.5–8 Å, see Supplementary Fig. 3), suggest that they correspond to two β-CD geometries on the surface.

The truncated cone shape of β-CD with its two distinct faces allows for two orientations on the surface where all seven glucose units are equally visible by STM and nc-AFM. In one case, β-CD is adsorbed on the surface via its secondary face, the primary face with its narrow rim pointing upwards (primary face up or p↑ orientation). In the other case, the secondary face with the wide rim faces the probe, while the primary face points towards the surface (secondary face up or s↑). Accordingly, based on the different diameters observed, the star-shaped objects are assigned to p↑, while the larger ones to s↑ orientation. The assignment is further confirmed by simulated nc-AFM

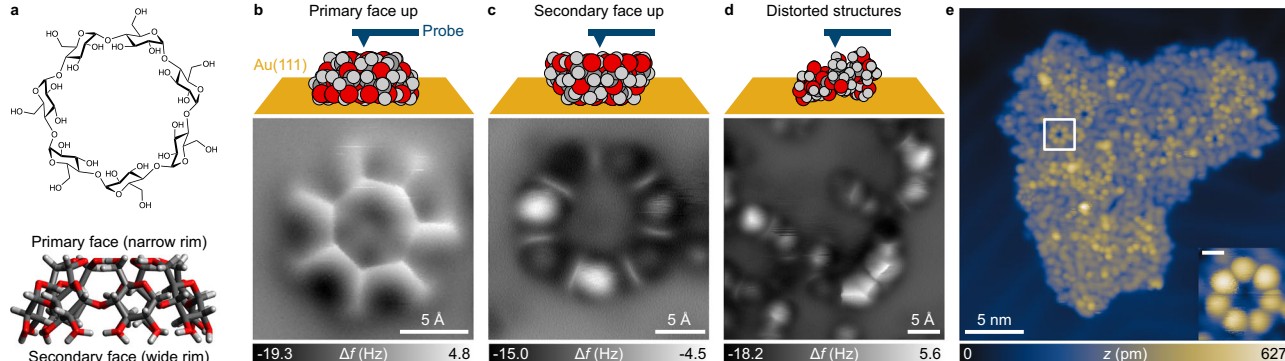

**Fig. 1 | Structure and real-space images of β-cyclodextrin molecules by low-temperature STM and nc-AFM. a** Chemical formula of β-CD and side-view of a 3D-model highlighting the molecule's truncated cone shape. **b–d** Nc-AFM images of β-CD on Au(111), recorded at constant height using frequency modulation detection. Representative examples are shown for the primary (**b**) and secondary face-up geometries (**c**), as well as for distorted molecules (**d**). **e** STM overview showing an irregular island formed upon electrospray deposition of β-CD onto Au(111) (bias = 1 V, tunneling current = 1 pA). The macrocycles stand out as doughnut-shaped structures from a lower background of fragments; the insert shows a higher resolution STM image of a single macrocycle (scale bar in the insert is 5 Å).

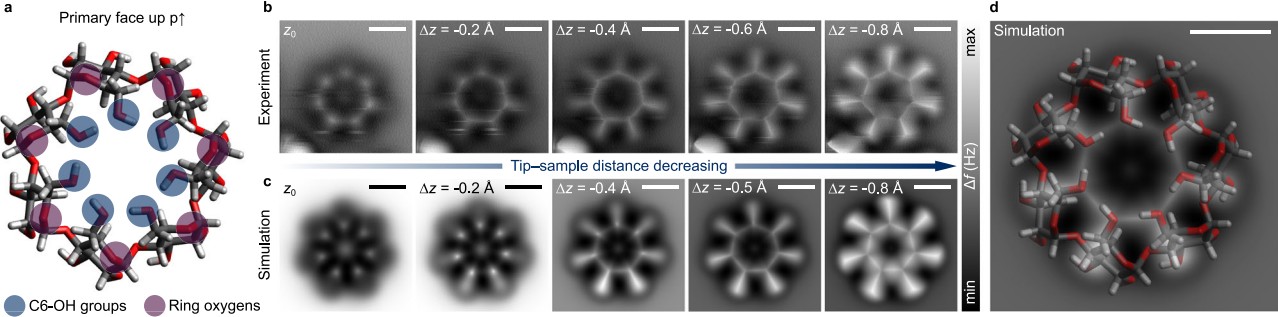

**Fig. 2 | Comparison of experimental and simulated nc-AFM images of the β-cyclodextrin primary face. a** Atomic model of β-CD, representing the lowest-energy conformer at the DFT PBE level of theory, viewed from its primary face. More prominent functional groups are highlighted in color for ease of discussion. **b** Experimental and **c** simulated nc-AFM images of β-CD in the p↑ geometry, the tip-sample distance decreasing from left to right. In the experiment, $z_0$ is defined as

+250 pm relative to an initial STM setpoint above the molecular ring with the feedback at 1 V and 3 pA; $\Delta z$ designates the difference in tip-sample distance relative to $z_0$. **d** Overlay of atomic model and corresponding simulated image of β-CD in the p↑ geometry; the exact atomic groups giving rise to the various features can be readily identified. All scale bars are 5 Å.

images of both geometries (discussed further in the next section). Additional STM and nc-AFM images of single β-CD molecules in p↑ and s↑ orientation are shown in Supplementary Fig. 9.

The third frequently observed object corresponds to distorted β-CD molecules (Fig. 1d). In this case, only a few of the round features observed in molecules in s↑ orientation can be resolved. Upon comparison with simulated nc-AFM images (see Supplementary Fig. 5), it becomes evident that these correspond to bent β-CD molecules, where only outward-facing features are visible in the constant-height nc-AFM images. Molecular dynamics (MD) simulations revealed that such distortion arises from the rotation of two α−1,4-glycosidic bonds, breaking the symmetry of the β-CD molecules (for more details see Supplementary Note 4 and Supplementary Fig. 4). This rotation causes two adjacent glucose units to twist, with their glucopyranose rings stacked on the surface. The remaining five glucose units remain in the original conformation. This conformational substate of the β-CD molecule was found to be a minor population in simulations in a solvent environment without including the surface. However, it is significantly stabilized when the β-CD molecule is adsorbed on the surface in the s↑ orientation (discussed in more detail in Supplementary Note 4.2).

The nc-AFM images were recorded in constant-height mode using frequency modulation detection. In this detection mode, more positive $\Delta f$ frequency shifts (depicted in bright gray) reflect enhanced repulsive forces between the CO molecule attached to the tip and the molecule on the surface[24,25]. The repulsion arises mainly from the steric Pauli repulsion at close tip-sample distances and alternatively from the electrostatic interaction. Thus, in the first approximation, we may interpret bright areas as regions of higher electron density within the molecules, often indicating the location of atoms and chemical bonds[26]. Dark areas, in contrast, correspond to regions with attractive interaction driven by long-range van der Waals and, alternatively, electrostatic interactions dominating over short-range steric repulsion. Considering these contrast mechanisms in nc-AFM, some tentative structural assignments may be made based on the experimental micrographs. In p↑, for instance, the bright borders of the heptagon, which outline a sharp ridge on the potential energy surface above the molecule, indicate the presence of a closed H-bonding network along the narrow rim of β-CD. In the case of s↑, the seven round features and ribs must arise from corresponding atomic groups of the seven glucose building blocks, which very likely involve the secondary OH groups. However, unambiguous identification of the exact atoms giving rise to these features or determining the ring puckers of the glucopyranose units is not possible based solely on the singular constant-height images. The characteristics of nc-AFM images are dominated by forces that

change with the tip-sample distance. Thus, for more definite structural assignments, experimental micrographs acquired at different tip-sample distances were compared to simulated images for a set of low-energy structural candidates.

## Structural assignment—comparison of experiment and theory

The precise structural arrangement of different β-CD conformers from high-resolution nc-AFM imaging was elucidated through comparison with simulated nc-AFM images using the probe-particle (PP) model[24]. In essence, the probe particle (here a CO molecule) positioned at the tip apex demonstrates sensitivity to spatial fluctuations in the potential energy landscape of the molecule, arising from the interplay among Pauli, electrostatic, and van der Waals interactions[27–29]. At short distances between the tip and sample, the probe particle adjusts in response to the potential energy surface, generating a distinct submolecular contrast. For modeling the sample, a set of low-energy gas-phase β-CD conformers were generated by classical MD simulations and subsequently relaxed on the Au(111) surface in both p↑ and s↑ geometry. Equilibrated molecules on the Au(111) surface were then used for additional total energy density functional theory (DFT) calculations using FHI-aims[30].

Figure 2 compares theoretical and experimental nc-AFM images acquired for the p↑ geometry. The optimized atomic model of the conformer of β-CD using total energy DFT calculations, which gave the most accurate match between simulations and experimental images, is shown from the side of its primary face in Fig. 2a (atomic coordinates provided as Supplementary Data 1, 2). The most important functional groups are highlighted in the figure in color, and the numbering of atoms is explained in Supplementary Fig. 8. The structure in Fig. 2a exhibits $C_7$ symmetry with all glucopyranose units adopting a $^4C_1$ ring pucker. Along the narrow rim of the primary face, a closed, homodromic H-bonding network is formed by the seven primary C6-OH groups (OH···O distances at 1.79 Å), all serving both as donor and acceptor. In both experiment and simulation, the resulting ridges in the interaction potential energy surface above the molecule in p↑ orientation become visible at close tip-sample distances as the sharp borders of the central heptagon (Fig. 2b, c). The gradual emergence, size, and symmetry of the seven rays observed in experiments are also accurately reproduced in the simulations, along with the position and contrast of the darker regions. Overall, the agreement between experimental and simulated nc-AFM images is highly accurate for the p↑ geometry. Overlay of the atomic model with a corresponding simulated image for p↑ also enables direct visual assignment (Fig. 2d). As mentioned above, the structural elements underlying the central heptagon are the primary C6-OH groups that form a closed network of H-bonds. This also suggests that β-CD may be present as neutral

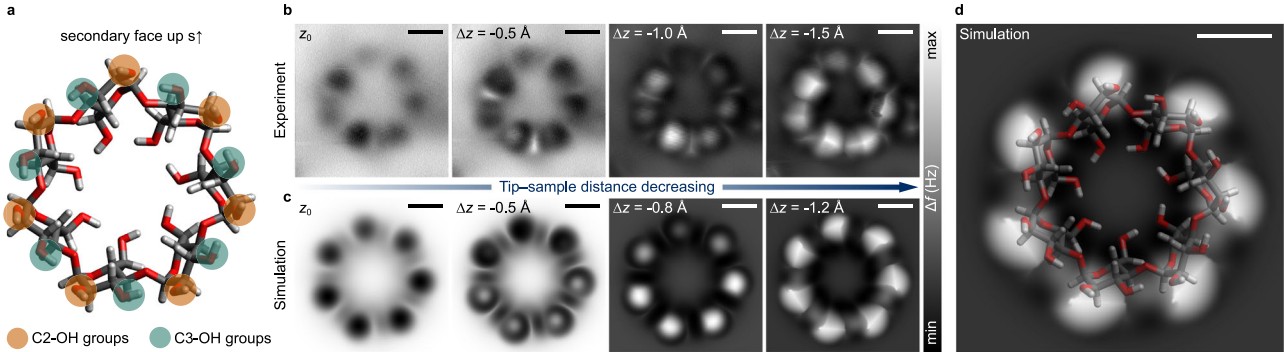

**Fig. 3 | Comparison of experimental and simulated nc-AFM images of the β-cyclodextrin secondary face. a** Atomic model of β-CD, representing the lowest-energy conformer at the DFT PBE level of theory, viewed from its secondary face. More prominent functional groups are highlighted in color. **b** Experimental and **c** simulated (bottom) nc-AFM images of β-CD in the s↑ geometry, the tip-sample distance decreasing from left to right. In the experiment, $z_0$ is defined as +190 pm relative to an initial STM setpoint above the molecular ring with the feedback at 0.5 V and 1 pA; Δz designates the difference in tip-sample distance relative to $z_0$. **d** Overlay of atomic model and corresponding simulated image of β-CD in the s↑ geometry. All scale bars are 5 Å.

species on the Au(111) surface following electrospray deposition, as coordination to small metal ions—potentially present in the electro-sprayed solutions—generally disrupts the regular intramolecular H-bonding pattern in isolated cyclodextrins[31]. The seven rays can be assigned mainly to a protruding hydrogen atom of methylene bridges that connect the C6-OH groups with the six-membered pyranose rings of the glucose units (see optimized DFT structure in Supplementary Data 1, 2). The ring oxygens located an atomic layer below, and the glycosidic oxygens, buried even deeper, are not directly visible in the images. These results show the superior resolution and additional structural information provided by nc-AFM as compared to conven-tional STM, where such assignments would be impossible due to lack of high-resolution, sub-monomer detail.

A similar comparison of experiment and simulation for the s↑ geometry is shown in Fig. 3. Here, the structural elements most acces-sible to the tip are the secondary OH groups, highlighted in color in Fig. 3a. These groups point anti-clockwise when seen from the secondary face, forming H-bonds in a pairwise manner along the wide rim where the C3-OH oxygens (O3) serve as donors and the C2-OH oxygens (O2) as acceptors. Simulated images for s↑ are also in excellent agreement with their experimental counterparts (Fig. 3b, c). The central cavity, the round features, and the narrow ribs observed experimentally are all correctly reproduced by theory at different tip-sample distances. By overlaying the atomic model on the corresponding simulated images, the seven round features and the seven ribs can be assigned to the C2-OH and C3-OH groups, respectively. Overall, the agreement between experiment and simulation is very convincing for both adsorption geometries, enabling a confident assignment of β-CD's atomic structure.

It needs mentioning that the β-CD molecules in s↑ orientation appear slightly distorted in the experimental images. Variations in the brightness of the round features and ribs indicate that, upon adsorp-tion via its primary face, β-CD can undergo slight deformation that leads to differences in the height of the secondary OH groups (typically within the fraction of an Å). While comparable variations in con-formation may prevent ensemble-averaged imaging methods from obtaining high-resolution structural information, nc-AFM allows for the imaging of single molecules in real space, thereby revealing the structural heterogeneity present in the sample.

Besides enabling the assignment of β-CD's atomic structure, simulations also helped provide a rationale behind the observed con-trast in the nc-AFM images. We found improved agreement between theoretical and experimental nc-AFM images when the electrostatic interaction between tip and sample was taken into account. The impact of electrostatics on the simulated images is shown in Fig. 4 for the s↑ orientation. The influence of electrostatics is also recognizable

for p↑ (see Supplementary Fig. 7), but is much more prominent in the case of the s↑ geometry. The greater impact of the electrostatic interaction on the latter reflects the stronger polar character of the secondary face. This results mainly from differences in the orientation and bonding of the C2-OH and C3-OH groups, which also contribute to their different contrast. The hydrogens of the C2-OH groups in the atomic model point away from the plane of the O2 atoms, protruding slightly from the secondary face. The hydrogens in the C3-OH groups, on the other hand, are located in the plane of the O2 atoms, with which they engage in H-bonds. Thus, the hydrogen atoms in the C2-OH groups are located somewhat closer to the plane of the tip. As the O–H bonds are highly polar, the difference in the orientation of the C2-OH and C3-OH groups leads to an alternating pattern of positive and negative electrostatic potential values above these moieties (see the electrostatic potential maps in Fig. 4e, j). The resulting difference in the electrostatic interaction, combined with variations in steric repulsion, is largely responsible for the different contrast of the two sets of sec-ondary OH groups. These findings highlight the sensitivity of nc-AFM to small differences in the orientation and chemical environment of otherwise very similar functional groups within a complex glycan structure.

Supplementary Fig. 6 shows simulated nc-AFM images for other stable and symmetric β-CD conformers from MD simulations, which display a different orientation of the hydroxy groups. However, these simulated images show significantly different sub-molecular contrast compared to the experiments, which further demonstrates the power of nc-AFM to identify atomic structural differences, in particular molecular conformations. Considering these findings, it is tempting to envisage similar sensitivity in identifying conformers and isomeric species from other glycan classes by nc-AFM, or even other biomole-cules of comparable size. As a method capable of providing atomic-level structural detail for complex, three-dimensional biomolecular species on the single-molecule level, as demonstrated herein, nc-AFM, in combination with ESD, has the potential to overcome many of the key challenges associated with glycan analysis, complement estab-lished analytical methods, and significantly improve our under-standing of the structure of glycans and other biomolecules.

## Discussion

This work demonstrates the high-resolution imaging of single glycan molecules using nc-AFM with CO-functionalized probes, providing atomic-level structural information on β-CD, a complex biomolecule with a distinctive three-dimensional shape. Compared to conventional STM methods, nc-AFM displays superior spatial resolution that reveals the internal structure of the monosaccharide building blocks.

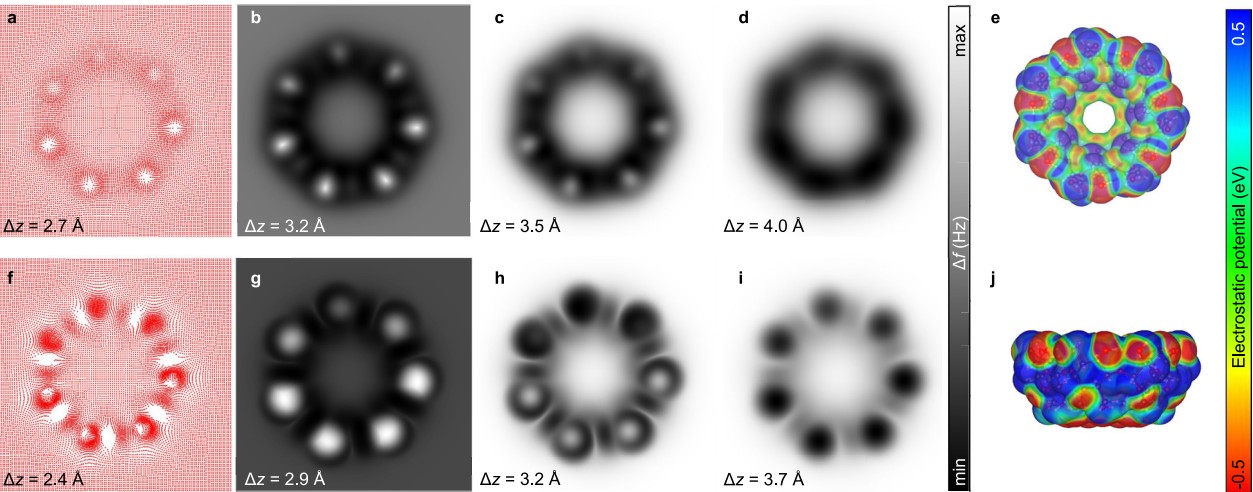

**Fig. 4 | Nc-AFM simulations and the electrostatic potential of β-cyclodextrin with its secondary face upwards.** The lateral relaxations of CO-like probe particles at tip-sample distances corresponding to the closest approach of the probe during the oscillation acquiring nc-AFM images shown in b and g are presented as red dots in panels **a** and **f**, respectively. Panels **b**–**d** hold the nc-AFM simulations without including the electrostatic potential, and sections **g**–**i** contain the simulations that include the electrostatic interaction between probe and sample. Panels **e**, **j** include the top (viewed from the secondary face) and side views of β-CD's electrostatic potential map, respectively. The $\Delta z$ values define the distance between the outermost atom of the molecule and the probe particle along the z-axis (perpendicular to the surface plane).

Individual OH groups on both faces of the molecule are resolved with atomic-scale detail, and the location of intramolecular H-bonds could be unambiguously determined from the atomically resolved nc-AFM images. By combining nc-AFM experiments with first-principles DFT simulations, the strong dependence of the imaging contrast on minor variations in the position and chemical environment of the secondary OH groups could be rationalized, allowing the unambiguous distinction and assignment of these key structural elements. The strong agreement between experiment and theory, taking into account the different adsorption geometries of β-CD, allowed for the confident assignment of the molecule's atomic structure.

The structure of intact β-CD on the weakly interacting Au(111) surface displays 7-fold symmetry. Every OH group in the molecule participates in intramolecular H-bonds: the C6-OH groups form a closed homodromic H-bonding network on the primary face, while the secondary OH groups are H-bonded in a pairwise manner. This structure, determined in the absence of solvent in UHV, is significantly different from the high-resolution models obtained via X-ray crystallography, where co-crystallized solvent molecules disrupt the intramolecular H-bonds of β-CD, altering its intrinsic conformation. Nc-AFM also reveals the different adsorption geometries of β-CD, as well as small structural variations between individual molecules on the surface. Thus, the present work provides novel insight into the structure β-CD that has so far not been attainable by established imaging methods.

This study represents the first application of high-resolution nc-AFM for glycan analysis and serves as a significant step toward extending this technique beyond the analysis of conjugated planar molecules. Our results may also facilitate further nc-AFM studies exploring more complex glycans from both synthetic and natural sources. Determining conformations, distinguishing isomers, and assigning linkage position and configuration in unknown structures are key challenges where nc-AFM has the potential to complement established mass spectrometric and spectroscopic methods. The combination of nc-AFM with electrospray ion-beam deposition (ESIBD)[32,33] to enable the soft landing of mass-selected ionic species would open further possibilities, including the analysis of complex glycan mixtures or studies on fragments generated in gas-phase reactions. Based on the findings presented herein, we anticipate that nc-AFM will play an important role in advancing our understanding of the structure of glycans and other complex biomolecules.

## Methods

### DFT calculations and simulation of nc-AFM images

DFT calculations for all free-standing molecules were performed using the FHI-aims package[30]. All geometry optimizations were carried out in the gas phase, using the generalized gradient approximation Perdew–Burke–Ernzerhof exchange–correlation functional[34]. In all the calculations, the light settings for the numerical atomic basis sets were employed. The Brillouin zone was sampled using the Gamma-point. Systems were allowed to relax until the Hellmann-Feynman forces reached values below $10^{-2}$ eV Å$^{-1}$.

Theoretical nc-AFM maps were calculated with the Probe-Particle package[24] for a CO-like tip, using the DFT-calculated structure and electric potential as an input. The AFM probe was modeled as the outermost atom of the metal tip to which a CO molecule (the probe particle) is covalently attached via its C atom. Simulated images were obtained by laterally scanning this probe with a step size of 0.1 Å above the sample. At each lateral position, the probe was approached towards the sample in steps of 0.1 Å and allowed to relax. Following relaxation at each step, the vertical component of the force exerted on the probe by the sample was calculated and subsequently converted into $\Delta f$ values. The simulated $\Delta f(x,y)$ maps were finally compared to those recorded experimentally.

### Electrospray deposition of β-cyclodextrin

For electrospray deposition (ESD), β-cyclodextrin (from Sigma-Aldrich, purity ≥97%) was dissolved in a 50:50 (v/v) mixture of water (Milli-Q®) and methanol (HPLC-grade, ≥99.9%) at a concentration of 1 mg mL$^{-1}$. The solution was then loaded into a gastight Hamilton syringe, connected to the emitter of a commercially available UHV4i ESD source[35] (Molecularspray Ltd.) through a PEEK capillary. To achieve stable spray conditions, the solution was pumped at a constant flow rate of 300 μL h$^{-1}$ using a syringe pump, with the emitter held at a potential of +2.3 kV. The molecules were deposited onto the (111) surface of an Au single crystal (MaTecK GmbH), held at room temperature in the UHV chamber of the microscope. During deposition, the pressure in the preparation chamber increased to the $10^{-7}$ mbar range from a base pressure of $10^{-9}$ mbar.

### Low-temperature STM and nc-AFM experiments

The experiments were conducted at a temperature of 4.2 K using a commercial STM/nc-AFM microscope from CreaTec Fischer & Co.

GmbH. Pt/Ir tips, sharpened by focused ion-beam (FIB), were utilized and were further cleaned and shaped by gentle indentations (≈1 nm) in the bare metallic substrate. The Au(111) substrate was prepared through repeated cycles of argon ion (Ar$^+$) sputtering at 1 keV, followed by annealing at ≈800 K.

STM topography was acquired in constant current mode with the bias voltage applied to the sample. Due to the low conductivity of the β-cyclodextrins, constant current images were typically performed with very low tunneling currents between 500 fA and 3 pA. In nc-AFM imaging, a qPlus sensor[36] (resonant frequency ≈30 kHz; stiffness ≈1800 N m$^{-1}$) was operated in frequency modulation mode with an oscillation amplitude of 50 pm.

The nc-AFM images were obtained in constant height mode using CO-terminated tips. The CO functionalization was performed by moving the tip in constant current to a CO molecule on the Au(111) surface with a setpoint of 20 pA tunneling current and 50 mV bias voltage. Once over the CO molecule, the STM feedback loop is opened, and the tip is approached towards the CO until a sudden drop in current and a corresponding change in frequency shift were observed.

The STM/nc-AFM images were processed using Gwyddion[37], WSxM[38], and SpmImage Tycoon[39] software.

### MD simulations
MD simulations were performed using the Gromacs 5.1.4 software package[40] with the Plumed extension[41]. β-CD was modeled with the GLYCAM04 force field[42–45]. The Au(111) surface, approximately 10 × 10 nm in size, was represented using a GolP force field, which explicitly accounts for induced polarization[46]. The Au(111) atoms were fixed in place, except for the rigid rod model of dipoles. The structure of β-CD was obtained from the Protein Data Bank (PDB ID: 3CGT). The β-CD molecule was positioned on the Au(111) surface with either its primary face up (p ↑) or secondary face up (s ↑). To enhance hydrogen bond stability on the primary face, a 1 kcal mol$^{-1}$ potential (referred to as HBfix) was applied between the heavy oxygen atoms[47]. Nonbonded interactions were truncated at 1 nm. Electrostatic interactions were treated using the Particle-Mesh Ewald method (PME) with the 1 nm cutoff for the real space term. The LINCS algorithm was used to constrain hydrogen atoms[48]. The equations of motion were integrated with a 2-fs time step. Each system was minimized, followed by a 5 ns NpT (in the case of solvent simulation) thermalization using a V-rescale thermostat[49] with a gradual temperature rise from 10 to 300 K and a coupling constant of 0.1 ps, and an isotropic Berendsen barostat[50] with a reference pressure of 1 bar and a coupling constant of 1.0 ps. The final production run was performed in the NVT ensemble for 1.1 μs (or for 100 ns for a set of 10 β-CDs and simulations with HBfix). Finally, the equilibrated molecules were cooled to 5 K in 2 ns to obtain low-energy gas-phase β-CD conformers for further ab initio simulations.

### Reporting summary
Further information on research design is available in the Nature Portfolio Reporting Summary linked to this article.

## Data availability
The data that support the findings of this study are available from Zenodo[51] and from the corresponding authors upon request. Optimized DFT structures are provided as Supplementary Data 1, 2.

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

## Acknowledgements

This work was supported by the Research Infrastructure NanoEnviCz, supported by the Ministry of Education, Youth and Sports of the Czech Republic (MEYS CR) under Project No. LM2023066 (B.T.); the DFG Walter Benjamin Program (Project GR 6150/1-1) (M.G.); the Czech Science Foundation 23-06781M (B.M., A.J.-M., and B.d.l.T.); the Czech Science Foundation 20-13692X (P.J., M.O., and S.E.); the UKRI Future Leaders Fellowship [MR/V02213X/1] (W.B.S.); the UKRI BBSRC (project BB/W017024/1) (W.B.S., S.R., and M.G.); the ERDF/ESF project TECHSCALE (No. CZ.02.01.01/00/22_008/0004587) (P.B., M.P., B.T., and P.K.). Computational resources provided by the e-INFRA CZ project (ID:90254), supported by MEYS CR, are acknowledged by M.O., S.E., and P.J.

## Author contributions

B.T., P.J., S.R., W.B.S., and M.G. conceived the research. B.M., M.G., A.J.-M., and B.T. performed the experiments and analyzed the data. S.E., M.O., and P.J. performed the DFT calculations and nc-AFM simulations. P.K., M.P., and P.B. performed the MD simulations. B.T., P.J., S.R., and P.B. supervised the project. All authors contributed to the writing of the manuscript. These authors contributed equally: M.G., B.M., and S.E.

## Competing interests

The authors declare no competing interests.
