## [Transparent Peer Review file · Nature Communications]

Atomically resolved imaging of the conformations and adsorption geometries of individual β -cyclodextrins with non-contact AFM

Corresponding Author: Dr Bruno de la Torre

Version 0:

Reviewer comments:

Reviewer #1

(Remarks to the Author)

The work shows atomic, bond-resolved resolution of cyclodextrin molecules obtained by non-contact atomic force microscopy (NC-AFM) with CO functionalized tips. It is a convincing demonstration, that such fragile molecules can be prepared by electrospray deposition for their atomic scale investigation by AFM. For that reason I support and suggest publication in nature communications. The data is of excellent quality and the combination of electrospray depositions with high resolution NC-AFM it an important milestone. Please consider the following comments:

Major:

1. The “imaging” of hydrogen bonds, mentioned in the abstract: “...the stabilizing H-bonds are imaged with atomic resolution...”, and in the conclusion: “ Individual OH groups and intramolecular H-bonding interactions – most prominently a closed H-bonding network of the C6-OH groups on the primary face – are resolved on the level of single bonds...”.

As the authors are probably aware of, there is a debate in the community, whether the contrast observed above hydrogen bonds stems from the hydrogen bonds or not, and thus whether hydrogen bonds are actually imaged by CO-tip AFM, or not. See, e.g. S. K. Hämäläinen et al. “Intermolecular contrast in atomic force microscopy images without intermolecular bonds,” Phys. Rev. Lett., vol. 113, p. 186102, 2014. I understand that contributing to this controversy is not in the scope of the paper. I think claiming that from the atomically resolved images the location of hydrogen bonds can be deduced, might be a safer statement, than the claim of direct imaging of hydrogen bonds. The latter might be disputed. Please consider rewording.

2. The reported positive frequency shifts (up to +17 Hz) and the contrast in the Df (frequency shift) scale (up to 30 Hz) are very large compared to other AFM works. Typically, the largest positive Df values that can be reached are a few Hz and the contrast in atom resolution is on the order of a few Hz in atomic resolution. Therefore, please provide the Df values for the individual AFM images shown in Fig. 2b and Fig. 3b, so the evolution of the values with distance can be observed. Optional, maybe comment of the large values. In part they are likely due the large corrugation of the sample. Could there be additional reasons? Also optional, show Df(z) spectra as in Fig. S3, but up to the distance where large pos. Df values are reached. Related question: Is there any contrast in the dissipation channel? In particular at the positions where large pos. Df values are measured? Please add that information about the dissipation.

3. It is not very clear, what is shown in Fig. 4 a and f (and Fig. S7a,f). What value is plotted and what do the colors mean? There seem to be different red tones. May be due to too low quality of the plot. What is the origin of the darker distorted grid, and what does it mean? The statement: “The lateral relaxations of CO-like probe particle at tip-sample distances corresponding to simulated nc-AFM images b and g are presented as red dots in panels a and f, respectively.” does not explain it very clear, as basically the entire images are mostly red. I assume positions of the oxygen atom of the tip are plotted. Is the value of the lateral relaxation presented by the red tone? Please provide the color scale of what is plotted and improve the description of these images and what can be seen in them.

4. It would be nice to clearly state and summarize what new information has been obtained with this study about the cyclodextrin molecules.

Minor:

5. p. 2. "However, STM images obtained with bare metal tips reveal the overall shape of individual monosaccharide building blocks convoluted with their adsorption configuration." Should the last word read "conformation" (and "convoluted" seems also not to be the right word here. Maybe: "...blocks, which also depends on their adsorption conformation."?)

6. p. 4. Please check, the term "free solvent simulations", to me it is not very clear what is meant (solvent-free?). It would be good to provide a reference for the statement of that sentence: "This conformational substate of the β -CD molecule was found to be a minor population in free solvent simulations."

7. In the context of Fig. S3, and the statement: "The height of an object (measured at a defined point) with respect to the Au(111) surface can be calculated as the difference between the two z_{min} values", I suggest citing: B. Schuler et al., "Adsorption Geometry Determination of Single Molecules by Atomic Force Microscopy," Phys. Rev. Lett., vol. 111, p. 106103, 2013. And instead of "calculated" maybe better: "measured".

8. Sample preparation: Was the sample annealed after the molecules had been deposited at room temperature, and before it had been cooled down for AFM measurements?

Signed report
Leo Gross

Reviewer #2

(Remarks to the Author)

In this manuscript, the authors combine electrospray deposition (ESD) in ultra-high vacuum with noncontact atomic force microscopy (nc-AFM) and theoretical calculations to unravel atomic-level structural information of β -cyclodextrin (β -CD). Two different conformations of β -CD were imaged by nc-AFM with high-resolution, and further elucidated through comparison with structural modeling and simulated nc-AFM images. By taking account the electrostatic tip-molecule interaction, they can distinguish and assign the different OH groups within single β -CD molecules. While nc-AFM has become a state-of-the-art characterization tool to determine the chemical structures of organic molecules, it represents a great challenge for nonplanar or 3D structures, especially large biomolecules. The data quality is high and this manuscript is well-written and logically structured. I would recommend the publication of the manuscript after the authors properly address the following concerns:

1. It seems that there miss the STM images of the two different β -CD conformers. Can they also be easily distinguished from STM images? And what configuration is the one shown in Fig. 1c? How are the electronic properties of the β -CD conformers? Will the local electronic states help to determine different chemical groups, especially when combined with AFM images?
2. In Fig. 3, the non-uniform ring in the nc-AFM image of the secondary face up β -CD was assigned to the slight deformation that leads to the height modulation of secondary OH groups. However, considering the dominant damage during deposition, the authors may also need to exclude other possibilities, such as OH dissociation.
3. In Fig. 4, the authors emphasized the important role played by the electrostatic potential in the nc-AFM imaging of the secondary face up β -CD, which presents the stronger polar character. I suggest the authors to use the Kelvin probe force microscopy (KPFM) to demonstrate the polar feature in the secondary face up β -CD in compared with the primary face up configuration. It would be more convincing for the argument.

Minor comments:

1. On page 5, the first occurrence of the abbreviation "MD" should provide the full spell "molecular dynamics (MD)".
2. In the z_0 definition, the tip hanging position before feedback off should be explicitly clarified with a real-space spot. The tip height should be significantly different on Au and on molecule with the same setpoint 1 V and 3 pA
3. Fig. S4 is not mentioned at all in the main text.

Reviewer #3

(Remarks to the Author)

Determining the structures of glycans is of great importance to understand their functionalities. In this manuscript, Grabarics et al. obtain high-resolution images of individual β -Cyclodextrins with Noncontact AFM. Although the electrospray deposition method and scanning tunneling microscopy had been used before to obtain the structures of biomolecules at the molecular level, this manuscript, to my best knowledge, achieves chemical-bond resolution with NC-AFM in the complex biomolecules for the first time. Also, the MD simulation and AFM simulation are in high quality.

I may recommend its publication in Nature Communications after addressing following questions:

1. The order of Fig. 1c and Fig. 1b could be reversed according to the description in the text.
2. It is better to have same scale bar for left and middle images in Fig. 1b, to compare their sizes in the same scale.
3. For the images of individual molecules, are they imaged in an island, or the molecules are moved out of an island? If it is the former case, the author should discuss how the local environment will change the charge distribution and bonding of individually measured molecules.

4. It is good to present the same area NC-AFM and STM images for individual molecules, to understand how the density of state distributes inside a molecule.
5. Regarding the intramolecular H-bonding network, it is good to present the charge difference map around the OH...O to indicate the formation of H-bond.
6. Why is the H-bonding network not observed in the S configuration in Fig. 3?
7. The authors mentioned the groups point anti-clockwise, as seen in the images. Is there some chiral features can be revealed?
8. It is not clear how Fig.4 a,f are plotted?
9. The authors mentioned in Fig. S6 a different conformation, which has not been observed in experiments. I am wondering is it due to the limited experimental results? So, totally how many images of individual molecules have been obtained? What is the ratio between the S and P configurations?

Reviewer #4

(Remarks to the Author)

Grabarics et al report their study using electrospray deposition in ultra-high vacuum combined with noncontact atomic force microscopy (nc-AFM) to investigate the conformation and atomic structure of individual β -cyclodextrins (β -CDs), a type of cyclic glucose oligomer. The high-resolution nc-AFM images revealed the different adsorption geometries and conformations of β -CDs, allowing for a much comprehensive assignment of their molecular structure. The study demonstrates the potential of nc-AFM for analyzing complex biomolecules like glycans at the atomic level. I find the approach is novel in determining the complicated structure of β Cyclodextrins, which is often challenging to obtain through other analytical techniques. The nc-AFM images obtained in this study demonstrate atomically spatial resolution, allowing for the visualization and assignment of individual functional groups within the β -CD molecules. The authors present experimental results with theoretical simulations to support their findings. This comprehensive analysis strengthens their conclusions regarding the structure and conformational flexibility of β -CDs. The manuscript is well organized and written. This article represents an important advancement in glycan analysis methodologies and has strong potential impact across various scientific disciplines. Therefore, I would like to recommend publication addressing some minor revisions suggested below.

1. While large-scale STM images were shown, I believe a zoom-in STM image could be helpful for the comparison in structural and electronic information, like in Fig.2a, even the focused issue is the AFM characterization.
2. The authors made a strong claim "Overall, the agreement between experiment and simulation is very convincing for both adsorption geometries, enabling a confident assignment of β -CD's atomic structure." While I found their assignments are reasonable, it is notice that the presence of OH groups and even C-H bonds often makes the complicated behaviours in AFM images. I suggest the authors give necessary discussion regarding such complications.

Version 1:

Reviewer comments:

Reviewer #1

(Remarks to the Author)

All points have been considered. I recommend publication.

Reviewer #2

(Remarks to the Author)

I believe the authors have adequately addressed the referees' concerns. Therefore I am pleased to recommend it for publication in Nature Communications.

Reviewer #3

(Remarks to the Author)

The authors have addressed my comments and revised the manuscript accordingly. I can recommend its publication now.

Reviewer #4

(Remarks to the Author)

While the authors adequately addressed my concern regarding the STM image, my second concern was not satisfactorily resolved. In their structural model of beta-CD, there are out-of-plane oriented C-H bonds that should have a height comparable to the O-H bond based on their provided structural models. For instance, in planar carbon nanoribbons, the out-

of-plane CH₂ and CH₃ groups often produce noticeable contrast in AFM images. It raises the question of how this situation might influence the AFM image of nearby O-H bonds. Such complexities should be clarified to convincingly assign beta-CD's atomic structure, yet this aspect is completely overlooked. Consequently, I cannot accept their claimed "convincing assignment of beta-CD's atomic structure" in its current form. Therefore, I cannot recommend it for publication until this issue has been adequately addressed.

Version 2:

Reviewer comments:

Reviewer #4

(Remarks to the Author)

The authors have address my concern, and have revised the manuscript appropriately. I recommend its publication.

Real-Space Imaging of the Conformation and Atomic Structure of Individual
 β -Cyclodextrins with Noncontact AFM

Manuscript # NCOMMS-24-37922-T

Referee comments are in italic font, our answers to them are in blue.

The changes are highlighted in yellow in the revised manuscript.

Reviewer #1

The work shows atomic, bond-resolved resolution of cyclodextrin molecules obtained by non-contact atomic force microscopy (NC-AFM) with CO functionalized tips. It is a convincing demonstration, that such fragile molecules can be prepared by electrospray deposition for their atomic scale investigation by AFM. For that reason I support and suggest publication in nature communications. The data is of excellent quality and the combination of electrospray depositions with high resolution NC-AFM it an important milestone.

We thank the referee very much for the time devoted to our manuscript, the very high appreciation of our work supporting and suggesting publication, and his insightful comments, which have been replied to point-to-point below, altogether re-enforcing the quality of our manuscript.

Please consider the following comments:

Major points

1. The “imaging” of hydrogen bonds, mentioned in the abstract: “...the stabilizing H-bonds are imaged with atomic resolution...”, and in the conclusion: “ Individual OH groups and intramolecular H-bonding interactions – most prominently a closed H-bonding network of the C6-OH groups on the primary face – are resolved on the level of single bonds...”.

As the authors are probably aware of, there is a debate in the community, whether the contrast observed above hydrogen bonds stems from the hydrogen bonds or not, and thus whether hydrogen bonds are actually imaged by CO-tip AFM, or not. See, e.g. S. K. Hämmäläinen et al. “Intermolecular contrast in atomic force microscopy images without intermolecular bonds,” Phys. Rev. Lett., vol. 113, p. 186102, 2014. I understand that contributing to this controversy is not in the scope of the paper.

I think claiming that from the atomically resolved images the location of hydrogen bonds can be deduced, might be a safer statement, than the claim of direct imaging of hydrogen bonds. The latter might be disputed. Please consider rewording.

We thank the reviewer for pointing out this inconsistency. We rephrased the respective parts of the text accordingly.

Abstract: "The position of individual hydroxy groups and the location of the stabilizing intramolecular H-bonds are deduced from atomically resolved images..."

Conclusion: "Individual OH groups on both faces of the molecule are resolved with atomic-scale detail, and the location of intramolecular H-bonds could be unambiguously determined from the atomically resolved nc-AFM images."

2. *The reported positive frequency shifts (up to +17 Hz) and the contrast in the Df (frequency shift) scale (up to 30 Hz) are very large compared to other AFM works. Typically, the largest positive Df values that can be reached are a few Hz and the contrast in atom resolution is on the order of a few Hz in atomic resolution. Therefore, please provide the Df values for the individual AFM images shown in Fig. 2b and Fig. 3b, so the evolution of the values with distance can be observed. Optional, maybe comment of the large values. In part they are likely due the large corrugation of the sample. Could there be additional reasons? Also optional, show Df(z) spectra as in Fig. S3, but up to the distance where large pos. Df values are reached. Related question: Is there any contrast in the dissipation channel? In particular at the positions where large pos. Df values are measured? Please add that information about the dissipation.*

We appreciate the reviewer's attention to detail regarding the reported values in some of our images. After careful review, we have identified that there was indeed an issue with the offsets of the total range of values in the analysing programs (Gwyddion and WSxM) that affected these values. We have since corrected this by re-analysing the raw data using their native acquisition software (Scan Viewer of Nanonis).

The images, once corrected, display conventional df values for the attractive regime between a minimum of around ~18 Hz to a maximum value for the repulsive regime of around +5 Hz. These values, updated thanks to the accurate observation of the referee, reinforce our thesis that the electrostatics between the tip and the molecule is responsible for a substantial part of the contrast of the AFM images.

Regarding Figures 2 and 3, we have decided to maintain the current min/max color bars for ease of visualization. These figures do not contain specific values that need correction. However, to ensure transparency and reproducibility, we will include the original raw data for all the figures in the final version of the paper, where among other channels, the excitation channel is available for all the images. In this channel (Attached Fig. 1R below), no evidence of dissipative processes is noted, and therefore,

all the abnormally large values can be explained in terms of the above-mentioned reasons.

Figure 1R. Example of constant height Frequency shift and Excitation channels obtained with a CO-functionalized tip of one β -CD molecule displaying the secondary face up.

3. It is not very clear, what is shown in Fig. 4 a and f (and Fig. S7a,f). What value is plotted and what do the colors mean? There seem to be different red tones. May be due to too low quality of the plot. What is the origin of the darker distorted grid, and what does it mean? The statement: “The lateral relaxations of CO-like probe particle at tip-sample distances corresponding to simulated nc-AFM images b and g are presented as red dots in panels a and f, respectively.” does not explain it very clear, as basically the entire images are mostly red. I assume positions of the oxygen atom of the tip are plotted. Is the value of the lateral relaxation presented by the red tone? Please provide the color scale of what is plotted and improve the description of these images and what can be seen in them.

The red dots indicate the lateral position of the probe particle at a given tip-sample distance on the grid, where the force field acting on the probe particle is calculated to reconstruct the simulated nc-AFM image. At larger tip-sample distances, where no lateral relaxation of the probe particle occurs, the grid of red dots appears uniform. At closer tip-sample distances, the positions of the red dots shift due to the lateral relaxation caused by tip-sample interaction.

Below the grid of red dots, the corresponding vertical relaxation of the probe particle along the z-axis is shown in a black-and-white color scale for a given tip-sample distance. In the revised version of the manuscript, we have decided to remove the color map of the vertical relaxation and only display the lateral relaxation of the probe particle, represented by the grid. We now intentionally show the lateral relaxation at the closest point of the probe's oscillation, which has the most significant impact on the AFM image. The figure caption has been modified accordingly. For the referee's convenience, we include the updated Figure 4 here:

Fig. 4 | Simulated nc-AFM simulations and the electrostatic potential of β -cyclodextrin with its secondary face upwards. The lateral relaxations of CO-like probe particle at tip-sample distances corresponding to the closest approach of the probe during the oscillation acquiring nc-AFM images shown in b and g are presented as red dots in panels a and f, respectively. Panels b–d hold the nc-AFM simulations without including the electrostatic potential, and sections g–i contain the simulations that include the electrostatic interaction between probe and sample. Panels e and j include the top (viewed from the secondary face) and side views of β -CD's electrostatic potential map, respectively. The Δz values define the distance between the outermost atom of the molecule and the probe particle along the z-axis (perpendicular to the surface plane).

4. *It would be nice to clearly state and summarize what new information has been obtained with this study about the cyclodextrin molecules.*

We appreciate the comment of the reviewer highlighting that the novelty of our study regarding the cyclodextrin molecules is not clear enough. Specifically, we resolved the atomic structure of BCD of molecules in a dry environment (no solvent), which is significantly different than that obtained by X-ray crystallography, where co-crystallized solvent molecules disrupt the intramolecular H-bonds of β -CD, altering its intrinsic conformation. In this regard, we have extended the Conclusion section of the manuscript to summarize the new information obtained about β -cyclodextrin in this study. Specifically, it now reads as follows:

“The structure of intact β -CD on the weakly interacting Au(111) surface displays 7-fold symmetry. Every OH group in the molecule participates in intramolecular H-bonds: the C6-OH groups form a closed homodromic H-bonding network on the

primary face, while the secondary OH groups are H-bonded in a pairwise manner. This structure, determined in the absence of solvent in UHV, is significantly different from the high-resolution models obtained via X-ray crystallography, where co-crystallized solvent molecules disrupt the intramolecular H-bonds of β -CD, altering its intrinsic conformation. Nc-AFM also reveals the different adsorption geometries of β -CD, as well as small structural variations between individual molecules on the surface. Thus, the present work provides novel insight into the structure β -CD that has so far not been attainable by established imaging methods.”

Minor points

5. p. 2. *“However, STM images obtained with bare metal tips reveal the overall shape of individual monosaccharide building blocks convoluted with their adsorption configuration.” Should the last word read “conformation” (and “convoluted” seems also not to be the right word here. Maybe: “...blocks, which also depends on their adsorption conformation.”?)*

“Configuration” has been corrected to “conformation”, and the rest of the sentence has also been rephrased accordingly.

“However, STM images obtained with bare metal tips reveal the overall shape of individual monosaccharide building blocks, which also depends on their adsorption conformation, while internal structural details at the atomic level remain unresolved.”

6. p. 4. *Please check, the term “free solvent simulations”, to me it is not very clear what is meant (solvent-free?). It would be good to provide a reference for the statement of that sentence: “This conformational substate of the β -CD molecule was found to be a minor population in free solvent simulations.”*

We acknowledge the reviewer for raising this question. We agree that the term free solvent simulation might be confusing. Accordingly, with the reviewer’s suggestions, the term “free solvent simulation” has been changed, and a reference has been added to the respective part of the Supplementary (“Section 4 and Supplementary Fig. 4”). In the revised version of the manuscript now, it can be read as: “This conformational substate of the β -CD molecule was found to be a minor population in simulations in a solvent environment without including the surface. However, it is significantly stabilized when the β -CD molecule is adsorbed on the surface in the $s\uparrow$ orientation (discussed in more detail in the SI, Section 4.2).”

7. *In the context of Fig. S3, and the statement: “The height of an object (measured at a defined point) with respect to the Au(111) surface can be calculated as the difference between the two*

zmin values”, I suggest citing: B. Schuler et al., “Adsorption Geometry Determination of Single Molecules by Atomic Force Microscopy,” Phys. Rev. Lett., vol. 111, p. 106103, 2013. And instead of “calculated” maybe better: “measured”.

We appreciate the suggestion of the referee. The text has been updated accordingly, and the suggested reference added to the revised version of the manuscript (Ref. 1 of Supplementary).

8. *Sample preparation: Was the sample annealed after the molecules had been deposited at room temperature, and before it had been cooled down for AFM measurements?*

The samples were not annealed between deposition and AFM analysis. However, it is interesting to comment that, in another set of experiments, a gentle annealing (100 °C for a few seconds) following the first trial deposition led to no significant changes. This scenario of being unaffected is compatible with the results from other groups using a similar electrospray setup for depositing DNA on Au(111) in UHV (DOI: 10.1038/s41467-019-08531-4).

Reviewer #2

In this manuscript, the authors combine electrospray deposition (ESD) in ultra-high vacuum with noncontact atomic force microscopy (nc-AFM) and theoretical calculations to unravel atomic-level structural information of β -cyclodextrin (β -CD). Two different conformations of β -CD were imaged by nc-AFM with high-resolution, and further elucidated through comparison with structural modeling and simulated nc-AFM images. By taking account the electrostatic tip-molecule interaction, they can distinguish and assign the different OH groups within single β -CD molecules. While nc-AFM has become a state-of-the-art characterization tool to determine the chemical structures of organic molecules, it represents a great challenge for nonplanar or 3D structures, especially large biomolecules. The data quality is high and this manuscript is well-written and logically structured. I would recommend the publication of the manuscript after the authors properly address the following concerns:

We thank the referee for the time devoted to our manuscript, the high appreciation of our work and his/her suggestion of publication after point-to-point revision to his/her comments, which has improved the quality of the manuscript.

Major points:

1. *It seems that there miss the STM images of the two different β -CD conformers. Can they also be easily distinguished from STM images? And what configuration is the one shown in Fig. 1c? How are the electronic properties of the β -CD conformers? Will the local electronic states help to determine different chemical groups, especially when combined with AFM images?*

We would like to acknowledge in this answer the interest of Reviewers 2, 3, and 4 for the STM topography images highlighting the interest of visualizing single β -CD molecules. Examples of STM images for each of the two different β -CD conformers have been added to the Supplementary Information (Supplementary Fig. 9) along with a constant height nc-AFM image to help the visualization of the structure. Based on STM images with CO-tips the two conformers can be distinguished, but beyond this distinction, little structural or electronic information can be extracted from the STM images. We would like to address here that glycans, such as the ones presented in our manuscript, are highly 3D, twisted objects, relatively tall (up to around 0.8 nm), and low-conductive (e.g. cellulose, starch, glucose, etc). Considering all these attributes, STM is heavily limited in exploring such systems with atomic accuracy as all the imaging in STM, either constant current or constant height, happens within the electronic band gap of the glycan.

2. *In Fig. 3, the non-uniform ring in the nc-AFM image of the secondary face up β -CD was assigned to the slight deformation that leads to the height modulation of secondary OH groups. However, considering the dominant damage during deposition, the authors may also need to exclude other possibilities, such as OH dissociation.*

In electrospray deposition the kinetic energy on impact is defined by the expansion into vacuum at the exit of the transfer capillary, beyond this point no additional forces that could accelerate the molecule are active (sample is grounded during deposition process in our system). In the expansion the molecule (β -CD) is moving with the carrier gas (mostly N_2) and hence the velocity will be approximately the sound velocity for air (450 m/s). For a molecule of around 1000 Da this amounts to an energy of approx. 1eV. At this collision energy we can safely assume soft-landing deposition, i.e. the adsorption of the molecule without breaking covalent bonds. This is confirmed by many experiments with ESD and ESIBD, where in the energy range of 0-5 eV fragmentation upon deposition has not been observed (<https://doi.org/10.1103/PhysRevLett.126.056001>, <https://doi.org/10.1038/s41586-020-2362-1>, <https://doi.org/10.1002/anie.201901340>). Also note, that the image charge potential is likely in the same range, which generally is not a problem in deposition. Lately, the perfect agreement between experimental and calculated nc-AFM images reinforces our assignment.

To make this point clearer, we modified the sentence: “The β -CDs are surrounded by a lower bed of smaller fragments (shown in Supplementary Fig. 2 in more detail), which most probably stem from the dissociation of the macrocycles during ESD, where electrochemical phenomena in ESD and collisions with the surface could both contribute to fragmentation.” Has been edited in the revised version of the manuscript which now reads as follow: “The intact β -CDs are surrounded by a lower bed that consists of distorted macrocycles (and potentially some smaller fragments), discussed later in more detail (additional images of the structures forming the bed are shown in Supplementary Fig. 2).”

3. In Fig. 4, the authors emphasized the important role played by the electrostatic potential in the nc-AFM imaging of the secondary face up β -CD, which presents the stronger polar character. I suggest the authors to use the Kelvin probe force microscopy (KPFM) to demonstrate the polar feature in the secondary face up β -CD in compared with the primary face up configuration. It would be more convincing for the argument.

We appreciate the referee's suggestion to employ Kelvin Probe Force Microscopy (KPFM) for investigating the polar features of β -CD configurations. Indeed, KPFM is a powerful, but time expensive, technique that has proven highly effective in probing local electrostatic distributions in various systems. However, applying it to our specific β -CD configurations presents significant challenges that we believe are important to address.

The complex 3D structure of β -CD molecules, with OH groups in varying spatial orientations, makes direct KPFM measurements and interpretations non-trivial. Moreover, the subtle height differences between the two conformations, coupled with their distinct charge distributions (7 vs. 14 exposed OH groups), create a complex electrostatic landscape between the molecules and CO-functionalized tip's apex that might not be easily resolved by KPFM. As some of us have already commented in previous works (DOI: 10.1038/s41467-023-40593-3), but also in other works performed by others (DOI: 10.1103/PhysRevLett.115.076101, 10.1103/PhysRevB.90.155455) the effect of tip-distance in the development of contrast in KPFM is critical and even subtle differences of height can contribute to dramatic changes of the contrast (We refer the Reviewer to the Supplementary materials of 10.1103/PhysRevLett.115.076101 for a stark example of change of the contrast).

Recent works by Ying Jiang's group (DOI:10.1126/science.abo0823, DOI:10.1038/s41586-024-07427-8) on water interactions with metals and insulators demonstrate that even in apparently simple systems involving OH groups of water molecules, the electrostatics of the system present a rich and complex playground

mediating all interactions between water molecules within the same plane in a 2D ices. This complexity is further amplified in our β -CD system.

Given these considerations, we believe that a direct 1-to-1 comparison between the two β -CD conformations using KPFM would be challenging and potentially inconclusive. Our current approach using nc-AFM and theoretical calculations provides a more suitable method for this specific study.

We appreciate the referee's suggestion and agree that a dedicated KPFM study of glycans could offer valuable insights, however, by itself, the charge distribution within glycans already qualifies as being its own line of research, and we believe is beyond the scope of our manuscript.

Minor comments

1. *On page 5, the first occurrence of the abbreviation "MD" should provide the full spell "molecular dynamics (MD)".*

We agree, the "MD" abbreviation is now introduced following the first full spelling of "molecular dynamics" (highlighted in yellow).

2. *In the z_0 definition, the tip hanging position before feedback off should be explicitly clarified with a real-space spot. The tip height should be significantly different on Au and on molecule with the same setpoint 1 V and 3 pA*

We thank the referee for pointing this out. Indeed, the tip height is significantly different on Au and on molecules. The mentioned set point height corresponds to the case where the tip stands above the molecular ring when the feedback is opened before constant height images are recorded. The information has been added to the captions of Figs. 2 (Primary) and 3 (Secondary) in the revised version of the manuscript, which now reads as follows: "In the experiment z_0 is defined as +250/+190 pm relative to an initial STM setpoint above the molecular ring with the feedback at..."

3. *Fig. S4 is not mentioned at all in the main text.*

We thank the reviewer for pointing it out. A reference to Fig. S4 has been added to the text. The added text is: "Molecular dynamics (MD) simulations revealed that such distortion arises from the rotation of two α -1,4-glycosidic bonds, breaking the symmetry of the β -CD molecules (for more details, see the SI, Section 4 and Supplementary Fig. 4)."

Reviewer #3

Determining the structures of glycans is of great importance to understand their functionalities. In this manuscript, Grabarics et al. obtain high-resolution images of individual β -Cyclodextrins with Noncontact AFM. Although the electrospray deposition method and scanning tunneling microscopy had been used before to obtain the structures of biomolecules at the molecular level, this manuscript, to my best knowledge, achieves chemical-bond resolution with NC-AFM in the complex biomolecules for the first time. Also, the MD simulation and AFM simulation are in high quality. I may recommend its publication in Nature Communications after addressing following questions:

We thank the referee for the time devoted to our manuscript and the positive evaluation of our work, recommending publication with minor revisions. We have addressed their comments point-by-point below, which have helped improve the manuscript.

1. *The order of Fig. 1c and Fig. 1b could be reversed according to the description in the text.*

We prefer to keep the figure as it is, as we believe the layout does not hinder the understanding of the scientific content.

2. *It is better to have same scale bar for left and middle images in Fig. 1b, to compare their sizes in the same scale.*

We appreciate the referee's attention to detail regarding the scale bars in Fig. 1b. We confirm that all scale bars in this figure are indeed uniformly set to 5 Å. This consistent scaling was intentionally chosen to facilitate direct and accurate size comparisons across all images.

Furthermore, we would like to draw attention to the main text, where detailed information about the sizes of the observed objects is already provided. Specifically, we state that the primary face has an outside diameter of 12.6 ± 0.5 Å, while the secondary face has 15.6 ± 0.1 Å across.

3. *For the images of individual molecules, are they imaged in an island, or the molecules are moved out of an island? If it is the former case, the author should discuss how the local*

environment will change the charge distribution and bonding of individually measured molecules.

We appreciate the Reviewer's question regarding the imaging environment of the individual molecules. To clarify, the molecules were imaged within their native island formations and were not manipulated prior to imaging. In fact, tip-induced lateral manipulations of B-CD can induce distortions even at the low temperature of the experiments, making it difficult to study the preferred molecular conformations.

However, our simulations were indeed carried out on isolated molecules on the surface. The excellent agreement between these simulated images and our experimental observations strongly suggests that the conformation and intramolecular noncovalent interactions of the molecules are minimally affected, if at all, by their surroundings in the islands.

It's important to note that while the molecules aggregate in islands, there are no covalent bonds between them. The aggregation is primarily driven by weak dispersive interactions between molecular surfaces. The size and number of molecules in these islands do promote this aggregation, but the intermolecular forces are weak enough that they do not significantly alter the individual molecular conformations or charge distributions.

This consistency between isolated simulations and in-island observations indicates that the local environment within the islands does not substantially change the charge distribution or bonding of individually measured molecules. The dominant intramolecular interactions appear to be preserved, allowing for an accurate comparison between experimental and theoretical results.

4. It is good to present the same area NC-AFM and STM images for individual molecules, to understand how the density of state distributes inside a molecule.

We thank the Reviewer for raising the question of how the molecules look in STM. We have prepared a new figure (Supplementary Fig. 9) displaying constant current topography and constant height ncAFM with CO-functionalized tip for the primary and secondary faces. We kindly address the referee to our answer for question 1 of Referee 2.

5. Regarding the intramolecular H-bonding network, it is good to present the charge difference map around the OH...O to indicate the formation of H-bond.

In this particular case of intramolecular hydrogen bonding, calculating the charge difference map between the OH–O forming the hydrogen bond is not a straightforward task. In the case of intermolecular hydrogen bonds, we can easily divide the system into two independent stand-alone molecular species, for which the charge density is well-defined. Consequently, the differential charge can be easily obtained by subtracting the charge density of individual molecules from the charge density of the whole system. However, in our case, hydrogen bonds are intramolecular. Therefore, we cannot easily calculate charge densities between the OH–O, which would require complicated partitioning of the cyclodextrin macrocycle itself, introducing spurious charge redistribution due to breakage and subsequent passivation of covalent bonds.

6. *Why is the H-bonding network not observed in the S configuration in Fig. 3?*

This is an interesting point raised by the reviewer. The difference in nc-AFM contrast arises from the highly three-dimensional structure of the secondary face. Unlike the primary face, which is planar, the secondary face features protruding H atoms from OH groups, as can be observed in the side view of the β -CD molecule (Figure 2R below). This structural distinction leads to the observed variation in nc-AFM contrast between the two faces due to the high tip-sample distance dependence of the nc-AFM imaging technique.

Figure 2R. Side view of the optimized atomic structure of a β -CD molecule obtained from DFT calculations.

7. *The authors mentioned the groups point anti-clockwise, as seen in the images. Is there some chiral features can be revealed?*

We thank the reviewer for this question, it is indeed a very interesting point. β -Cyclodextrin is a chiral molecule which comes in enantiopure form, meaning there is only one enantiomer (consisting of D-glucose units) on the surface.

As mentioned, on the secondary face of the molecule the OH groups all point in an anti-clockwise direction. This is the most stable conformer of β -cyclodextrin, lower in energy than the conformer where the secondary OH groups point in the clockwise direction (the primary OH groups being clockwise in both).

At close inspection in the original simulated nc-AFM image of β -cyclodextrin (Fig. 3Ra), we may observe some directional features that reflect the anti-clockwise orientation of the OH groups of the secondary face in a very close tip-molecule distance. For instance, in Fig. 3Ra, the white part of the ridge between the lobes is always clockwise from the lobes, while the dark part is anti-clockwise. If we mirror this image onto a plane perpendicular to the surface plane, the resulting mirror image (Fig. 3Rb) will not be superimposable with the original by translation or rotation around an axis perpendicular to the surface plane. For example, the white part of the ridge between the lobes will be anti-clockwise from the lobes, while the dark part is clockwise (exactly the opposite as in the original). Thus, we can say that the directionality of the OH groups on the secondary face is visible in the simulations.

We must emphasize however that these subtle features are not unambiguously resolved in our experimental data. While our experimental results are consistent with the simulation, we cannot definitively resolve the dark and bright sides of the ridges or the directional 'deformation' of the lobes experimentally. Our conclusion that the OH groups point in anti-clockwise direction is therefore based on energetic considerations: in the most stable conformer the secondary OH groups are anti-clockwise, the conformers with clockwise secondary OH groups are higher in energy.

Figure 3R. Simulated nc-AFM image of the secondary face of β -cyclodextrin with the OH groups pointing in anti-clockwise direction (a), and the mirror image obtained by reflection of the nc-AFM image onto a plane perpendicular to the surface plane (b).

Reviewer #4

Grabarics et al report their study using electrospray deposition in ultra-high vacuum combined with noncontact atomic force microscopy (nc-AFM) to investigate the conformation and atomic structure of individual β -cyclodextrins (β -CDs), a type of

cyclic glucose oligomer. The high-resolution nc-AFM images revealed the different adsorption geometries and conformations of β -CDs, allowing for a much comprehensive assignment of their molecular structure. The study demonstrates the potential of nc-AFM for analyzing complex biomolecules like glycans at the atomic level. I find the approach is novel in determining the complicated structure of β Cyclodextrins, which is often challenging to obtain through other analytical techniques. The nc-AFM images obtained in this study demonstrate atomically spatial resolution, allowing for the visualization and assignment of individual functional groups within the β -CD molecules. The authors present experimental results with theoretical simulations to support their findings. This comprehensive analysis strengthens their conclusions regarding the structure and conformational flexibility of β -CDs. The manuscript is well organized and written. This article represents an important advancement in glycan analysis methodologies and has strong potential impact across various scientific disciplines. Therefore, I would like to recommend publication addressing some minor revisions suggested below.

We thank the referee for their time and positive assessment of our manuscript, recommending publication with minor revisions. We have addressed each of their comments below, contributing to the manuscript's improvement.

1. *While large-scale STM images were shown, I believe a zoom-in STM image could be helpful for the comparison in structural and electronic information, like in Fig.2a, even the focused issue is the AFM characterization.*

We thank Reviewer 4 for noting the question of how the molecules look in STM in concordance with the interest of Reviewer 2 and 3. We have prepared a new figure (Supplementary Fig. 9) displaying constant current topography and constant height nc-AFM with CO-functionalized tip for the primary and secondary faces. We kindly address the referee to our answer for question 1 of the Referee #2.

2. *The authors made a strong claim "Overall, the agreement between experiment and simulation is very convincing for both adsorption geometries, enabling a confident assignment of β -CD's atomic structure." While I found their assignments are reasonable, it is notice that the presence of OH groups and even C-H bonds often makes the complicated behaviours in AFM images. I suggest the authors give necessary discussion regarding such complications.*

Unfortunately, we do not fully understand the referee's comments regarding the complicated behavior of OH groups and C-H bonds in AFM images. The contrast in nc-AFM images with CO-tips is extensively discussed in existing literature and is

briefly addressed in our study. The contrast mechanisms, even in the presence of OH groups and C-H bonds, are now well understood.

Real-Space Imaging of the Conformation and Atomic Structure of Individual
 β -Cyclodextrins with Noncontact AFM

Manuscript # NCOMMS-24-37922-T

Referee comments are in italic font, our answers to them are in blue.

The changes are highlighted in yellow in the revised manuscript.

Reviewer #1:

All points have been considered. I recommend publication.

We thank Professor Leo Gross for supporting the publication of our manuscript in its present form.

Reviewer #2:

I believe the authors have adequately addressed the referees' concerns. Therefore I am pleased to recommend it for publication in Nature Communications.

We thank the Reviewer for his/her for recommending the publication of our manuscript.

Reviewer #3:

The authors have addressed my comments and revised the manuscript accordingly. I can recommend its publication now.

We thank the Reviewer for his/her recommendation for publication in its actual form.

Reviewer #4:

While the authors adequately addressed my concern regarding the STM image, my second concern was not satisfactorily resolved. In their structural model of beta-CD, there are out-of-plane oriented C-H bonds that should have a height comparable to the O-H bond based on their provided structural models. For instance, in planar carbon nanoribbons, the out-of-plane CH₂ and CH₃ groups often produce noticeable contrast in AFM images. It raises the question of how this situation might influence the AFM image of nearby O-H bonds. Such complexities should be clarified to convincingly assign beta-CD's atomic structure, yet this aspect is completely overlooked. Consequently, I cannot accept their claimed "convincing assignment of

beta-CD's atomic structure" in its current form. Therefore, I cannot recommend it for publication until this issue has been adequately addressed.

We thank Reviewer 4 for the time dedicated to evaluating our work and regret that we could not fully convince him/her in our previous response. In this response, we have paid special attention to this last main concern, regarding the atomic structure of β -CD molecules and their AFM visualization.

Indeed, at the interface of β -CD model there are out-of-plane C-H bonds, as seen in Figure R1, which contribute substantially to the high-resolution AFM contrast. Namely, the protruding H_1 hydrogen of methylene bridge together with the oxygen atom (denoted as O in Figure R1) is responsible for the radial lines observed in nc-AFM images, see Figure 2 and Supplementary Figure 7, as already discussed in the text. Note that another hydrogen H_2 of methylene bridge is located 0.5 Angstrom below -OH group. Consequently, it cannot be resolved by CO-tip at a tip-sample distance where the experimental images were taken.

Moreover, we would like to stress that slightly different atomic arrangements of -OH and -CH₂ groups presented in Supplementary Figure 6 provide completely different high-resolution nc-AFM contrast. This demonstrates the large sensibility to discriminate the particular structural arrangement and enables us to make a convincing assignment of β -CD's atomic structure.

To make the discussion clearer, we modified the manuscript in the following way (changes marked in yellow):

"The seven rays can be assigned mainly to a protruding hydrogen atom of methylene bridges that connect the C6-OH groups with the six-membered pyranose rings of the glucose units (see optimized DFT structure in Supplementary Material)."

We also decided to include the DFT-optimized structure of cyclodextrin in XYZ format to facilitate readers a better understanding of the structure as well as the possibility of performing nc-AFM simulations themselves, e.g., using the web interface of the probe particle AFM simulator, which is available here: <http://ppr.fzu.cz/>.

Figure R1. Detailed zoom-in of the side view of the atomic structure of the β -CD molecule.